# Relationships between Math Skills, Motor Skills, Physical Activity, and Obesity in Typically Developing Preschool Children

**DOI:** 10.3390/bs13121000

**Published:** 2023-12-07

**Authors:** Pedro Flores, Eduarda Coelho, Isabel Mourão-Carvalhal, Pedro Forte

**Affiliations:** 1CI-ISCE, Higher Institute of Education and Sciences of the Douro, 4560-708 Penafiel, Portugal; pedro.flores@iscedouro.pt; 2Department of Sports, University of Trás-os-Montes and Alto Douro, 5000-801 Vila Real, Portugal; 3Research Center in Sports, Health and Human Development, 6201-001 Covilhã, Portugal; 4Department of Sports, Instituto Politécnico de Bragança, 5300-252 Bragança, Portugal

**Keywords:** math skills, motor skills, physical activity, obesity

## Abstract

There is evidence of a relationship between motor and cognitive development. The literature has shown that of all the motor skills, fine motor skills are those that contribute most to mathematical performance in preschool children. As this is a sensitive period in the development of motor skills, low levels of physical activity in this period can compromise their development and contribute to weight gain and obesity. The aim of this study was therefore to analyze the relationship between mathematical and motor skills, physical activity levels, and obesity. The sample consisted of 62 preschool children (32 males) with an average age of 4.63 ± 0.81. The Weschler preschool and primary scale of intelligence—revised arithmetic test was used to assess mathematical skills. The tests to assess fine motor skills were the “Adapted Threading Beads Test” and the “Adapted Visuomotor Integration Test”. The movement assessment battery for children-2, band 1, “Aiming & Catching”, and “Balance” tests were used to assess gross motor skills. Levels of physical activity were assessed using the “Preschool-age physical activity questionnaire” and obesity using the body mass index. The results indicated that only the fine motor skills of visuomotor integration were included in the multiple linear regression model (F < 0.001; r = 0.464; R^2^ = 0.215; *p* < 0.001), with the exclusion of gross motor skills, physical activity levels, and obesity levels. Thus, it was concluded that mathematical skills were only directly and significantly influenced by visuomotor integration. However, visuomotor integration was positively and significantly associated with gross motor skills (r = 0.269; *p* < 0.05) and not with levels of physical activity and obesity. Thus, gross motor skills could contribute to improving visuomotor integration directly and consequently mathematical skills indirectly. The results of this study suggest that the implementation of structured physical activity programs can contribute to mathematical performance.

## 1. Introduction

The early childhood period is a critical phase in the physical, cognitive, social, and emotional development of a child [1]. During this period, there is a significant and intense reorganization of the central nervous system [2,3,4], which is strongly influenced by the child’s environmental context [3,5]. It is a time of great opportunities but also of significant vulnerability to negative influences [2].

During this period, the majority of a child’s time is spent in family and educational environments [6,7]. Therefore, in addition to the family, schools have the responsibility to promote the holistic development of children in their early school years [6,7,8].

The literature has shown that in early childhood, a child’s physical, cognitive, social, and emotional development is influenced by motor development [9,10,11]. It is during this period, before the age of six, that significant motor skill development occurs [12,13].

From a developmental system’s perspective, children’s motor skills are influenced by the complex interplay of biological, social, and environmental factors [14,15]. As a result, there are various theories of development, and numerous authors have explored the relationships between cognitive and motor development [16].

Currently, there is a clear connection between the brain areas involved in motor skills (primarily the cerebellum) and cognitive skills (primarily the prefrontal cortex) [17,18,19]. The development of both skills occurs simultaneously and rapidly in the early years of life [18,20,21]. In this regard, motor skills acquired early in life are related to cognitive abilities from childhood [22] to adulthood [10,23].

Recent studies have shown that motor skills influence academic performance in the early years [24,25,26,27,28,29,30,31] and are described as one of the criteria for school readiness [32]. However, motor skills do not develop naturally, they need to be learned, practiced, and assimilated [33,34,35]. Therefore, in early childhood, there appears to be a close relationship between motor skill development and physical activity (PA) [36,37]. A more active child may have better motor skills [31,38,39,40], with a reduced risk of obesity [36,41,42]. On the other hand, the presence of obesity contributes to physical inactivity and consequently may lead to underdeveloped motor skills [36,43,44]. In this sense, well-developed motor skills in early childhood facilitate children’s engagement in PA [36,45,46,47,48,49] and, as a result, contribute to the prevention of obesity [36,42].

Motor skills refer to the underlying internal pathways responsible for moving the body through space, as well as the cognitive processes that give rise to these movements [50,51]. Traditionally, they are categorized into two main groups (Figure 1): gross motor skills (GMSs) and fine motor skills (FMSs) [52,53,54,55,56].

GMSs primarily involve movements generated by large muscle groups. They encompass locomotor skills, which entail body movements in space (such as walking, running, jumping, and sliding), postural or balance skills, which relate to the ability to maintain a controlled position or posture during a task (dynamic balance—maintaining position during activities that involve movement; or static balance—maintaining position in stationary tasks), and manipulative skills used to control objects through actions with hands or feet (e.g., grasping, striking, absorbing, lifting, etc.). These manipulative skills can be categorized as either propulsive (sending objects) or receptive (receiving objects) [57,58,59,60,61].

FMSs are characterized by movements performed by small muscle groups. The literature often subdivides fine motor skills into two categories. One is fine motor coordination (FMC), related to eye–hand coordination, also known as non-graphomotor ability, which involves manual dexterity, speed, and precision in tasks such as inserting, moving, and handling small objects with the fingers [62,63].

Another category of fine motor skills is visual–motor integration, known as visuomotor integration (VMI) or visuospatial integration, which relates to the organization of small hand and finger movements through the processing of visual and spatial stimuli, with a strong emphasis on synchronized movements between the hand and the eye. These tasks typically include activities like writing, drawing, copying shapes, letters, or other visual stimuli and can be referred to as graphomotor skills [62,63].

**Figure 1 behavsci-13-01000-f001:**
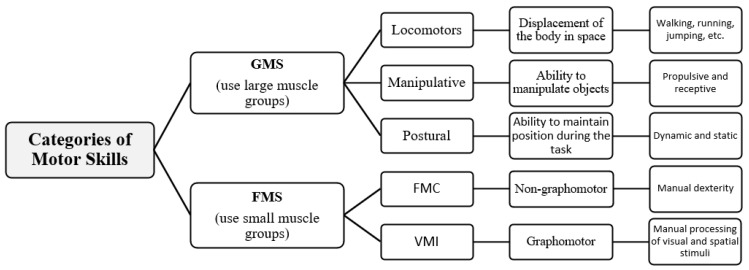
Summary of motor skills categories (adapted from Flores et al. [64]).

Since the development of motor skills in early childhood contributes to a greater involvement of children in the practice of PA, it is important to characterize and classify this concept.

PA is characterized by all voluntary bodily movements performed by skeletal muscles that require energy expenditure [65]. PA can be classified into structured PA (formal PA conducted under the guidance and direct instruction of an expert) and unstructured PA (informal PA performed spontaneously and freely by children through play activities). Structured PA aims to improve motor skills or specific physical fitness, while unstructured PA is more associated with recreational activities [66].

Regarding the frequency and intensity of PA, the World Health Organization (WHO) recommends that preschool children engage in at least 180 min of PA per day, of which 60 min should be of moderate to vigorous intensity (MVPA) [67].

Prolonged periods of low PA in children can lead to weight gain [67,68,69]. The primary mechanism leading to weight gain is an imbalance between energy consumption and energy expenditure [70].

The WHO [71] defines obesity as the abnormal or excessive accumulation of body fat that poses a risk to an individual’s health. Fat accumulation results from increased energy intake exceeding the energy expenditure, leading to a positive energy balance [72]. Energy consumption can be explained using the metabolic equivalent (MET). One MET (4.2 KJ) corresponds to 1 Kcal/kg body weight/hour, which is approximately equal to the resting metabolic rate [73]. When sitting (sedentary PA-SPA), only 1 to 1.5 METs are used; light PA (LPA) consumes between 1.5 and 3 METs; moderate PA consumes 3 to 6 METs; and vigorous PA consumes more than 6 METs [73,74]. In this context, low levels of PA are associated with obesity development in children and adolescents [75].

The literature has demonstrated the effects of motor skills and PA on cognitive and academic skills, showing a positive relationship between these skills in preschoolers [31,76,77,78,79,80,81]. On the other hand, it has been shown that obesity can compromise children’s participation in PA and negatively influence motor skills [36,43,44] and cognitive skills [82]. In this regard, early childhood motor skills can be the foundation for a more active lifestyle [83,84], as studies have demonstrated a positive association between motor skills and higher levels of PA [40,83,85], resulting in reduced obesity [36,42]. Therefore, higher levels of PA in early childhood are positively associated with many aspects of health, such as reduced obesity [86,87], development of motor skills [40,83,85], and cognitive skills [88,89,90].

Cognitive and academic skill performance primarily emphasizes two curriculum areas, literacy and mathematical performance [91,92]. These two areas are considered prerequisites for success in other disciplines and consequently for academic success [93]. Mathematics plays a prominent role in the school curriculum, and its development is considered a fundamental cognitive attribute. Successful early mathematical learning not only provides a structure for future learning [94] but also serves as an indicator of future academic and professional success [95].

Although the literature is more consistent in supporting a greater influence of FMSs on mathematical performance [29,64,96], some studies support positive and significant associations of GMSs on mathematical performance in preschool children [22,26,97]. In addition to motor skills, there is strong evidence that PA in young children is positively related to cognitive development [98] and specifically mathematical performance [99,100]. On the other hand, it seems that increased obesity leads to decreased PA [36,41], which contributes to a decrease in GMSs [38,39,40,101,102].

In view of the above, given that preschool age is a sensitive period for the development of motor skills and that low levels of PA in this period can compromise their development and contribute to weight gain and obesity, the following objectives were outlined: (1) to analyze the prevalence of difficulties shown by the children in mathematical skills, motor skills, compliance with weekly PA recommendations, and obesity levels, comparing the results with those in the literature, with the aim of justifying the fit or bias of the study variables; (2) to study the associations between all the variables with the aim of analyzing the direct relationships between them; (3) to analyze the direct influence of motor skills, PA levels, and obesity on mathematical skills.

## 2. Materials and Methods

### 2.1. Sample

All 115 children attending the preschool at the Vilela School Cluster (Paredes, Porto, Portugal) were invited to participate in the study. However, 5 children were excluded because they were unable to complete the assessments, and another 5 were excluded because they were born prematurely. Typically, among such children, the most common issues are related to both FMSs and GMSs [103,104], which negatively impact their academic performance [105,106].

Of the remaining 105 eligible children, 12 were absent for the FMS assessment, 7 were absent for the math skills assessment, 2 were absent for both assessments, 4 were part of a pilot study (preliminary validation of motor tests for assessing FMS), 10 children did not submit the questionnaire to assess their levels of PA, and 12 children had incomplete or improperly filled-out questionnaires. Thus, the final sample consisted of 62 children (32 males), with an average age of 4.63 ± 0.81 years, distributed across 5 classes (JIV1 to JIV5). In terms of age distribution, the 5-year-old age range predominated (33 children) (Table 1).

### 2.2. Instruments

#### 2.2.1. Diagnosis of Math Skills

The Weschler preschool and primary scale of intelligence—revised (WPPSI-R) [107] was revised and adapted for the Portuguese population [108], and given the objectives and characteristics of the study, it was the most appropriate scale and therefore the one used to diagnose mathematical skills. Given the scarcity of instruments for assessing math skills in preschool children in our country, this scale was chosen because it has been validated for the population in question, is the most widely used in national research at this level, does not require much training, and is easy to apply [108].

The WPPSI-R consists of two subtests, the achievement and the verbal. For this study, only the arithmetic test from the verbal subtest was used. This test assesses quantitative concepts, counting and problems, which are presented orally to the child. This test has 23 items divided into 3 parts: (i) quantitative skills, picture items (items 1 to 7); (ii) numerical skills, counting items (items 8 to 11); (iii) problem-solving skills, verbal items (items 12 to 23). The test should start at item 1. Although there is no time limit for the child to respond, after 15 s for items 1 to 11 and 30 s for items 12 to 23, a score of zero should be given, and the test should move on to the next item. One point should be awarded for each correct response (maximum score of 23 points), and the test should be stopped after 5 failures. Raw scores are converted into standardized scores, yielding a profile. Standardized scores range from 1 to 19, with a mean value of 10 and a standard deviation of 3 [107,108]. The classification assigned to the test based on standardized scores is as follows: 1–4, very inferior; 5–7, inferior; 8–12, average; 13–16, superior; 17–19, very superior.

#### 2.2.2. Diagnosis of Fine Motor Skills

To assess FMSs, two tests were used. To exclusively evaluate FMC, the adapted threading beads test (TBT-AD) was employed, and to assess visuomotor integration (VMI), the adapted visuomotor integration test (VMI-AD) was used [109]. These two tests were adapted to allow educators to assess FMSs by simultaneously administering them to the entire class, their students, in a classroom setting, within a short time frame, using minimal materials, and with straightforward interpretation of the results obtained from the administration [109].


*Adapted Threading Beads Test (TBT-AD)*


This test was adapted from the threading beads test in band 1 of the movement assessment battery for children-2 (MABC-2) [110] to allow it to be administered simultaneously in the classroom by educators to preschool pupils [109]. To this end, adaptations were made to the materials used and the application procedures. The main adaptation concerned the administration procedures, since the original test was only designed for the individual assessment of children [110]. In this way, the time spent on the assessment depends on the number of children being assessed, and when educators administer the test individually, they focus their attention on the child being assessed and may neglect others. Thus, the original test quantifies the time (in seconds) it takes to string a total of cubes on a string (6 cubes for 3- and 4-year-olds and 12 cubes for 5- and 6-year-olds) [110], while the adapted test quantifies the number of cubes strung on the string within specific time periods for each age group [109]. Thus, the following times have been proposed for TBT-AD according to age (years:months): 3:0–3:5, 26 s; 3:6–3:11, 23 s; 4:0–4:5, 21 s; 4:6–4:11, 17 s; 5 to 6:11, 24 s [109]. Based on the number of cubes strung on the string, a cut-off point was proposed to identify children without FMC disorders. Children aged 3:0–4:11 should thread at least 2 cubes, with the exception of children aged 4:0–4:5, who should thread 3 cubes. Children aged 5:0–5:11 must stick at least 5 cubes, and children aged 6:0–6:11 must stick at least 6 cubes [109]. However, depending on the age of the child and the number of cubes strung, a classification has been proposed for FVC: 0—severe disorder; 1—moderate disorder; 2—average; 3—good; 4—very good [109]. The degree of reliability of TBT-AD was analyzed using the intra-observer test–retest method, using intraclass correlation coefficients (ICCs) to assess its temporal stability. When the classroom group was assessed simultaneously, it obtained an ICC of 0.957 and individually an ICC of 0.924. This result indicates that TBT-AD has an excellent level of reliability, which allows it to be used for simultaneous administration in classroom groups, since its results remained stable in all evaluations, both group and individual [109]. With regard to predictive criterion validity, although TBT-AD was valid for explaining the results of the WPPSI-R arithmetic test (F = 6.949; *p* = 0.010; R^2^ = 7.8%), its correlation with this test was low (r = 0.280) [109].


*Adapted Visuomotor Integration Test (VMI-AD)*


This test was adapted from the reduced version of the Beery-Buktenica developmental test of visuomotor integration (VMI), which is suitable for preschool children [111]. Like the original test, the aim of the adapted visuomotor integration test (VMI-AD) is to accurately copy 15 geometric shapes. The main adaptation is related to the number of sheets used to administer the test, i.e., the original test consists of 7 sheets [111] and the VMI-AD test uses just one A4 sheet folded in half lengthways to form 4 pages, where the first page presents the test instructions and the following pages, 2, 3 and 4, present the 15 figures to copy (5 figures per page) [109]. The test lasts approximately 10 min; however, if a child has not completed it, the examiner should allow them to finish. One point is awarded for each shape copied correctly, for a total of 15 points. If the child fails to copy three consecutive shapes, the grading must end. In this regard, based on the standard score obtained, a performance profile was assigned: 40–67 points, very low; 68–82, low; 83–117, average; 118–132, high; 133–160, very high [111]. Based on the classification, the following points were assigned: zero points for very low performance; 1 point for low performance; 2 points for average performance; 3 points for high performance; 4 points for very high performance.

The degree of reliability of VMI-AD was analyzed using the intra-observer test–retest method with ICCs to assess its temporal stability. When the classroom group was assessed simultaneously, it achieved an ICC of 0.958, and individually, it achieved an ICC of 0.961. This result indicates that VMI-AD exhibits an excellent level of reliability, allowing it to be used for simultaneous classroom group administration, as its results remained stable across all assessments, both in group and individual settings [109]. With regard to predictive criterion validity, VMI-AD was valid for explaining the results of the WPPSI-R arithmetic test (score: F = 15.986; *p* < 0.001; classification: F = 12.300; *p* = 0.001), exerting an influence of 16.3% (score: R^2^ = 0.163) and 13.0% (classification: R^2^ = 0.130). These values showed a medium effect [112] and the value of the correlations a moderate effect [113].


*Scoring and Classification of Fine Motor Skills*


To calculate the score for FMSs, the standard scores of FMC and VMI were summed. To classify FMSs, the classifications of FMC and VMI were also summed, and the mean was calculated, rounded to the nearest decimal place. The following classification was assigned: 0 points, severe impairment; 1 point, moderate impairment; 2 points, average performance; 3 points, good FMSs; 4 points, excellent FMSs.

#### 2.2.3. Diagnosis of Gross Motor Skills

The GMSs were assessed using the MABC-2 [110], justified by the fact that this instrument is the most widely used to identify children with developmental coordination disorder [114,115,116,117]. Additionally, most of children’s learning occurs through manual exploration (manipulative, receptive, and propulsive skills), and balance (static and dynamic) is present in virtually all motor tests and tasks [110], while ball skills involve an important cultural aspect as they are part of children’s games and activities [118].

This instrument is divided into three bands: ages 3 to 6 years (band 1), 7 to 10 years (band 2), and 11 to 16 years (band 3). In this study, band 1 was used, including only the tests that assess manual ball skills (2 tests, aiming and catching) and balance (3 tests, 1 static and 2 dynamic). During the tests, the following behaviors should be recorded in addition to the task scores: “F” if the child does not perform the task correctly; “I” if the task is inappropriate for the child; “R” if the child refuses to perform the task.

The results were recorded on the “Age Range 1 Form (3–6 years)”. To obtain the scores for ball skills, the results of the “Bean Bag Catch and Throw” tests were summed and the respective standard scores, as proposed by the authors [110], were assigned. To obtain the scores for balance skills, the standard scores for unipedal balance (static) (preferred and non-preferred leg) were first summed and divided by 2 (average). Then, the standard scores for the 3 tests (unipedal balance–static, tiptoe walking, and jumping on mats—dynamic balance) were summed, and the respective standard score was assigned. According to the authors of the MABC-2 [110], the motor classification is as follows: red zone (motor disorder), standard score and percentile ≤ 5; yellow zone (risk of motor disorder), standard score between 6 and 7 and percentile between 6 and 15; green zone (no motor disorder), standard score > 7 and percentile ≥ 16.


*Manual Ball Skills (Aiming and Catching)*

*Bean Bag Catch (Aiming and Catching 1)*


Materials: bean bag and 2 mats; setup: a distance of 1.8 m between the mats. The examiner stands on one mat, and the child stands on the other mat facing the examiner; task: the examiner throws the bean bag to reach the level of the child’s outstretched hands (approximately between the waist and shoulders); trials: 10 attempts. Do not count if the examiner’s throw is poorly executed. If the child fails an attempt, the examiner should correct the error; record: number of attempts correctly executed.


*Throwing the Bean Bag (Aiming and Catching 2)*


Materials: bean bag and 2 floor mats (1 is the target); setup: 1.8 m distance between the mats; task: the child stands on the solid-color mat and throws the bean bag, trying to hit any part of the blue mat (the target); trials: 10 attempts. Number of attempts correctly executed without stepping off the mat while throwing noted.


*Balance Skills (Balance)*

*Unipedal Balance (Static)*


Materials: 1 mat and 1 stopwatch; setup: unobstructed space. Place the mat on the floor; task: the child must remain on one leg with their arms free on the mat for up to 30 s. If balance is maintained for 30 s, a second attempt is not required. No assistance can be given during the test; record: number of seconds, up to 30, that the child maintains balance without moving the supporting foot, touching the ground with the free foot, wrapping the free leg around the supporting leg, or gripping the free leg.


*Tiptoe Walking (Dynamic)*


Materials: colored tape; setup: mark a 4.5 m line on the ground with colored tape. The examiner should observe the child’s feet from the side to check if the heels touch the ground during the task; task: start the task with the child’s toe at the beginning of the line. Walk along the line with the heels raised without stepping off the line; trials: 2 attempts. If the child completes 15 steps without any mistakes, a second attempt is not needed. No assistance can be given during the test. Record: record the number of consecutive correct steps (maximum of 15). The child must not land outside the next mat or on the same mat, display an extreme loss of balance upon landing, touch the ground with their hands, or jump more than once on a mat.


*Jumping on Mats (Dynamic)*


Materials: 6 floor mats (3 solid-yellow, 2 solid-blue, and 1 target mat); setup: position the six mats side by side in a row, with the long sides touching and the colors alternating; task: the child starts with feet together on the first mat and jumps from mat to mat to the last one; trials: 2 attempts. If the child makes five continuous jumps without errors, a second attempt is not necessary. No assistance can be given during the test. Record: number of consecutive correct jumps (maximum of 5).

#### 2.2.4. Obesity Diagnosis

The body mass index (BMI) was the selected anthropometric indicator for assessing body composition, as it is the most widely used and accepted international predictor as an indirect measure of adiposity in children [119]. To perform this, the following variables needed to be collected: age, sex, height, and weight.

The BMI was calculated according to the formula: weight (kg)/[height (m)]^2^. The children’s age and sex were provided by the educators based on enrolment records.

Height was measured using a “Seca 213—Precision for health” stadiometer, with the child standing vertically, feet together, and arms extended alongside the body, with the head positioned so that the Frankfurt plane was parallel to the ground [120]. The measurement was recorded to the nearest millimeter (0.1 cm).

To assess weight, a Tanita digital scale (BC-730) was used. These materials were provided by the Instituto Superior de Ciências Educativas do Douro (ISCE Douro, Penafiel, Portugal), and the measurement was recorded to the nearest decigram (0.1 kg). The assessment of weight and height was consistently conducted under the same conditions and with the same equipment. The children were measured and weighed barefoot and wearing as little clothing as possible.

Both the European Childhood Obesity Group [121] and the International Pediatric Association [122] recommend using the cutoff points defined by the WHO for children from 0 to 5 years [123] and those aged 5 to 19 years [124] to standardize protocols and improve the quality of comparisons between different studies for the purpose of enhancing the quality of research and surveillance of the issue.

Thus, for the classification of BMI, percentiles for age were considered according to the criteria and cutoff points defined by the WHO [123,125,126]: underweight, percentile ≤ 5; normal weight, percentile between 5 and 85; pre-obesity, percentile > 85 and < 97; obesity, percentile ≥ 97.

#### 2.2.5. Physical Activity Assessment

To assess the children’s PA levels, the most commonly used objective methods include accelerometers, pedometers, and heart rate monitors. However, the use of these instruments is associated with several issues: the inability to assess aquatic activities, the absence of consensus regarding data cleaning and processing, and accelerometer cutoff points. Additionally, these instruments are relatively expensive for use in large populations [127]. In this context, subjective methods include tools like PA diaries and PA recall questionnaires, which are more practical and economically feasible for researchers to employ in large population studies [128,129].

Therefore, considering the nature and context of this study, the children’s PA level was assessed using the preschool-age physical activity questionnaire (Pre-PAQ) [130], the Portuguese version translated by Sancho [131]. This questionnaire represents a significant contribution for researchers to assess habitual PA and sedentary behavior over 3 days (1 weekday and 2 weekend days) in preschool-aged children [130]. Weekend days were included because activity routines at home tend to differ more on weekends than on weekdays [132,133]). The Pre-PAQ is a reliable and valid measurement tool for assessing various levels of PA in preschool-aged children [130,131,134,135].

The psychometric properties of the Portuguese version of the Pre-PAQ indicate that reliability values range from 0.402 to 0.938, demonstrating acceptable reliability for the translated version [131]. The Portuguese version of the Pre-PAQ [131] consists of 37 questions (Q1 to Q37). Considering the aim of this study, to collect information on children’s sedentary behaviors and PA, only questions Q32, Q36, and Q39 were selected. These questions include a list of typical preschool activities with “Yes” or “No” responses, and if “Yes”, the time the child spent on that activity.

Regarding question Q32, its purpose was to identify whether the child engages in any structured PA (physical education at school, swimming, dancing, soccer, ballet, etc.) inside and/or outside of kindergarten, specifying the activities and the time spent on them. Question Q36, focusing on the previous weekday (Monday to Friday), identified the types of activities the child engaged in, along with the time spent on each. Finally, question Q39, considering the previous weekend (Saturday and Sunday), identified the types of activities the child participated in, along with the time spent on each.

Question Q32 is related to the child’s participation in structured PA and the time spent on them. Questions Q36 and Q39 are related to the types of activities and the time spent on them by the child on a weekday (yesterday) and on the weekend, respectively. These questions encompass 22 types of activities. According to Cox et al. [136], the activities performed by children are categorized as follows: 1, 2, 3, 4, and 5, stationary/sedentary activities (SPAs); 6, 11, 15, 16, and 21, slow/light activities (LPAs); 7, 8, 9, 10, 12, 13, 14, 17, 18, 19, 20, and 22, moderate to vigorous activities (MVPAs).

Regarding time and intensity, the guidelines from the WHO [137] were followed, recommending at least an average of 60 min of moderate to vigorous PA daily.

### 2.3. Procedures

To collect all the data, the research project was first presented to the Director of the Agrupamento de Escolas de Vilela (Paredes, Porto, Portugal) and submitted for approval by the Pedagogical Council. After obtaining consent from the School Group, permission was sought from parents for their child’s participation in the research. Parents were informed about the study’s objectives, nature, methods, and all the tests the child would undergo, with an assurance that the collected data would remain confidential and used solely for research purposes. The informed, voluntary, and clear consent form for participation in the study was provided in writing and signed by parents before the research activities began.

The tests for evaluating FMSs and math skills were administered by a single investigator (intra-observer) to each of the classes simultaneously during the morning of 16 March 2023, following the respective protocols [109].

Regarding the WPPSI-R, it was individually administered in a designated space (following the protocol of the test) between 20 March and 24 March 2023, during the morning. The test administrator received instructions from a psychologist who is an expert in the field.

For the assessments of GMSs and BMI, these were conducted by students majoring in Sports Science at ISCE Douro between March 20 and March 24, in the morning. These students were instructed in February 2023 with theoretical presentations of the test protocols and subsequent practical experimentation.

Regarding the children’s PA data, the educators handed the Pre-PAQ to the respective guardians for completion between March 27 and March 31. Questionnaires with incomplete, incorrect, or missing data were excluded.

All procedures were in accordance with the Declaration of Helsinki for research involving human subjects.

For the analysis of results, in the initial phase, considering the classifications of the study variables, the children’s prevalence was described regarding: (1) performance on the WPPSI-R; (2) performance of FMSs (FMC and VMI); (3) performance of GMSs (ball skills and balance); (4) participation in structured PA and adherence to weekly PA recommendations; (5) the prevalence of obesity, considering the BMI classification.

In the second phase, the relationship between the study variables and their direct or indirect influence on the performance in the WPPSI-R test was analyzed.

### 2.4. Data Analysis

Descriptive statistics were employed to calculate the sample distribution, mean as a measure of central tendency, standard deviation as a measure of dispersion, and frequencies to describe the prevalence in the distributions of study variables, specifically performance on the WPPSI-R test, motor skills (FMSs—TBT-AD and VMI-AD; GMSs—MABC-2, aiming and catching, and balance), PA (structured PA and the number of days meeting weekly PA recommendations), and BMI classification.

Pearson’s correlation coefficient (r) was used to analyze the correlations between the study variables, and the following classifications were proposed: trivial (r ≤ 0.1), small (r = 0.1–0.3), moderate (r = 0.3–0.5), large (r = 0.5–0.7), very large (r = 0.7–0.9), and almost perfect (r ≥ 0.9) [113].

To analyze the influence of motor skills, PA, and BMI classification on mathematical performance, multivariate analysis was performed using the stepwise multiple linear regression (MLR) method. MLR examines how multiple independent variables relate to a dependent variable [138]. The MLR model can accurately suggest the relationships between variables and indicate the best effect of independent variables on the dependent variable (regression equation) [139]. This method is commonly used in educational research to measure the effects of explanatory variables on performance [140]. The F-test was used to assess the linearity of variable inclusion in the MLR model. The R test (Pearson) was used to examine the association between the variables for inclusion in the MLR model. The following classifications were proposed to investigate the influence between the variables (R^2^): values equal to 2% small effect; 13% medium effect; 26% as a large effect [112]. The standardized regression coefficients (β) and the non-standardized regression coefficients were used to analyze the influence that each variable represents in the MLR model. A significance level of *p* < 0.05 was set. The SPSS program, version 26.0, was used for the statistical analysis.

## 3. Results

### 3.1. Included Variables’ Prevalence

Regarding mathematical performance (Table 2), the majority of children did not exhibit difficulties in the arithmetic test of the WPPSI-R (75.8%). However, it was observed that 24.2% (15 children) displayed below-average performance, with only 17.7% (11 children) performing above average.

Table 3 presents the prevalence related to FMSs, where no students were observed to have significant difficulties or perform very well. Approximately 11.3% of students demonstrated some difficulties in their performance. Thus, the vast majority exhibited moderate performance (67.7%). Concerning the TBT-AD test performance, it is notable that none of the children exhibited severe disorders in their FMC. However, none achieved a very high performance, and a substantial number of children showed moderate disorders in their FMC (33.9%). In the VMI-AD test, similarly, none of the children demonstrated very high performance. Nevertheless, three children (4.8%) displayed very low performance, and 15 children displayed low performance (24.2%). These data reveal that approximately 34% of children have difficulties in their FMSs and 29% in their VMI.

Regarding GMSs, only six children (9.7%) showed a risk of motor disorder. In both ball skills (aiming and catching) and balance (static, dynamic, and total), the vast majority of children did not exhibit a risk of motor disorder (aiming and catching = 87.1%—54 children; balance static = 88.7%—55 children; balance dynamic and total = 91.9%—57 children). It was in the balance (dynamic and total) domain where the lowest number of children at risk was observed (4 children—6.5%) or with a motor disorder (1 child—1.6%). However, it is worth noting that six children (9.7%) showed motor disorder in ball skills and were at risk of motor disorder in balance static (9.7%) (Table 4).

Regarding PA (Table 5), the majority of children participate in structured PA (45 children—72.6%), with the practice of one structured PA session per week being prevalent (46.8%). However, in terms of meeting the weekly PA recommendations, only 27 children (43.5%) met the recommendations for daily intensity (a minimum of 60 min of MVPAs). Concerning PA intensity, the greatest amount of time was spent on SPAs (M = 328.95 ± 155.23) and the least on LPAs (M = 131.93 ± 96.39). The time spent on SPAs was significantly higher than the time spent on LPAs (*p* < 0.001) and similar to the time spent on MVPAs (*p* = 0.254).

As for BMI, its mean was 16.37 ± 2.22 (minimum = 11.12; maximum = 24.5), and the percentile mean was 59.62 ± 32.16 (minimum = 0.1; maximum = 99.8). Despite the majority (39 children—62.9%) being classified as having normal weight, it was observed that 9 (14.5%) children were in the pre-obese category and 9 others had obesity (Table 6).

### 3.2. Association between All Variables and Their Influence on Mathematical Skills

In the associations between all variables (Table 7) and concerning the performance in the WPPSI-R, only a significant moderate positive association was observed between the VMI-AD test (r = 0.464; *p* < 0.01) and the FMSs (r = 0.454; *p* < 0.01). The FMS result was obtained by summing the TBT-AD and VMI-AD tests. In this regard, the association of FMSs with the WPPSI-R was primarily influenced by the VMI-AD test, as the TBT-AD test exhibited a trivial negative association with the WPPSI-R (r = −0.045; *p* > 0.05). Moreover, the associations of the WPPSI-R with all other tests were trivial or small.

Regarding the result of FMSs, despite the positive influence of the TBT-AD test, this association was small and not significant (r = 0.177; *p* > 0.05). Thus, the outcome of FMSs was primarily influenced by the results of the VMI-AD test (r = 0.975; *p* < 0.001).

An important observation is that an increase in SPAs exhibited a significant negative association with the TBT-AD test performance (r = −0.267; *p* < 0.05) and a positive but not significant association with the VMI-AD test performance (r = 0.217; *p* > 0.05). Therefore, SPAs showed a positive association with the VMI-AD test and a negative association with the TBT-AD test.

Concerning the association between FMSs and GMSs, a small positive but non-significant association was found between these motor skills (r = 0.224; *p* > 0.05). However, this result was justified only by the VMI-AD test, as a significant positive association was observed only between this test and GMSs (r = 0.269; *p* < 0.05), but not with all GMS tests (VMI-AD—aiming and catching: r = 0.185; *p* > 0.05; VMI-AD—static balance: r = 0.121; *p* > 0.05; VMI-AD—dynamic balance: r = 0.160; *p* > 0.05; VMI-AD—total balance: r = 0.213; *p* > 0.05). On the other hand, the TBT-AD test showed small negative associations with GMSs (r = −0.186; *p* > 0.05), particularly with the aiming and catching test (r = −0.199; *p* > 0.05) and dynamic balance (r = −0.108; *p* > 0.05), and trivial associations with static balance (r = 0.057; *p* > 0.05) and total balance (r = −0.072; *p* > 0.05). These results suggest that the association between FMSs and GMSs was primarily influenced by the VMI-AD test.

Regarding GMSs, contrary to FMSs, all tests evaluating these skills contributed significantly to this association (aiming and catching: r = 0.765; *p* < 0.01; static balance: r = 0.450; *p* < 0.01; dynamic balance: r = 0.579; *p* < 0.01; total balance: r = 0.716; *p* < 0.01), with a similar contribution of aiming and catching and balance to GMS development. PA levels also had a small association with GMS, mainly SPAs (r = 0.252; *p* < 0.05) and LPAs (r = 0.204; *p* > 0.05), and not MVPAs (r = −0.110; *p* > 0.05). Additionally, an increase in BMI classification contributed slightly to decreased GMSs, as an increase in BMI classification had a negative association with GMSs (r = −0.216; *p* > 0.05) and their respective tests (aiming and catching: r = −0.188; *p* > 0.05; static balance: r = −0.050; *p* > 0.05; dynamic balance: r = −0.022; *p* > 0.05; total balance: r = −0.130; *p* > 0.05).

Regarding PA, children engaged in structured PA were significantly more likely to meet the weekly PA recommendations for a greater number of days (r = 0.272; *p* < 0.05). A significant association was also found between structured PA and PA levels, particularly with SPAs (r = 0.259; *p* < 0.05) and MVPAs (r = 0.193; *p* > 0.05), but not with LPAs (r = −0.171; *p* > 0.05). The association of structured PA with BMI classification was trivial (r = −0.030; *p* > 0.05). Concerning the days when children met the weekly PA recommendations, an increase was significantly associated with greater MVPAs (r = 0.764; *p* < 0.01) and not with SPAs (r = 0.181; *p* > 0.05) or AFL (r = 0.068; *p* > 0.05).

As for the increase in BMI classification, it had trivial and non-significant associations with other study variables, except for a slight and non-significant decrease in GMSs (r = −0.216; *p* > 0.05), negatively associated with all its components.

After analyzing the association between all variables, this study aimed to find the best multiple linear regression (MLR) model to explain the results of the WPPSI-R arithmetic test. Therefore, a multivariate analysis was conducted through MLR using the stepwise method to determine which independent variables of the study could directly or indirectly influence performance in the WPPSI-R arithmetic test.

Table 8 presents a summary of the MLR model to determine which independent variables should be included in the model, considering the WPPSI-R arithmetic test as the dependent variable. Table 9 shows the respective standardized regression coefficients (β) and unstandardized regression coefficients (Beta).

Based on the analysis of Table 8 and Table 9, it was observed that MLR selected only one model, including only the VMI-AD test. This test was the only one that demonstrated linearity in the F-test (F = 16.447; *p* < 0.001), showing a moderate positive association with the WPPSI-R (Beta = 0.464; T = 4.055; *p* < 0.001) and influencing it by 21.5% (R^2^ = 0.215). These results suggest that VMI-AD test scores can be used to explain the results of the WPPSI-R arithmetic test since, according to Polit and Beck’s criteria [112], these values are considered moderate (>13%) and close to a large effect size (≥26%).

The values recorded with MLR between R^2^-R^2^(Adj) were too high (0.013; 1.3%) to allow generalization to the population, since the values for the effect should be less than or equal to 0.004 (0.4%) of the variance.

Table 10 lists the tests that were excluded from MLR because they did not demonstrate significance for the effect (*p* > 0.05).

These results indicate that the CMF, GMSs, PA, and BMI classification did not directly influence math skills. Therefore, since the VMI-AD test was the only one to have a direct influence on the WPPSI-R arithmetic test, we sought to analyze whether any of the studied variables could indirectly influence the WPPSI-R arithmetic test through their influence on the VMI-AD test. For this purpose, we used the VMI-AD test as the dependent variable in MLR. The results showed that none of the variables were included in the model. These data suggested that FMC, GMSs, PA, and BMI classification did not directly influence VMI-AD. Thus, the WPPSI-R arithmetic test was indirectly influenced by FMC, GMSs, PA, and BMI classification.

## 4. Discussion

The main objective of this study was to study the relationships between math performance, motor skills, PA, and BMI classification. In view of the objective, it was considered important, in the first phase, to describe the prevalence of the difficulties shown by the children, in order to characterize the sample and compare the results with those in the literature to check if any variables could bias the study. Subsequently, to answer the main objective of the study, the associations between all the variables and their influence on mathematical performance were analyzed.

### 4.1. Prevalence Analysis

In terms of prevalence, regarding math performance, there was a prevalence of 24.2% of preschool children who showed difficulties in completing the WPPSI-R arithmetic test. The literature has shown very heterogeneous results regarding the mathematical difficulties experienced by preschool children [141], suggesting that these difficulties can vary between 10 and 30% [142,143]. The results of this study are in line with the literature, suggesting early diagnosis of mathematical skills, since the mathematical skills that children acquire in preschool are important for developing a conceptual understanding of math, predicting their performance up to eighth grade [144], and are a strong indicator of future academic and professional success [95].

Regarding FMSs, there was a prevalence of 11.3% of children with difficulties in these skills. As with the prevalence of math difficulties, the literature is also inconsistent regarding the prevalence of preschool children with FMS disorders [53,145]. Studies have demonstrated a variability in the prevalence of FMS disorders between 20% [146] and 25% [147]. However, other studies have found a lower prevalence, between 9.7% [148] and 11.7% [149]. Once again, the results of the present study are in line with the literature, where international research has shown that many preschool children have revealed difficulties in performing age-appropriate fine motor tasks [148,150]. Since low levels of performance in FMSs are associated with difficulties in learning mathematics [151], there is an urgent need to identify delays in these skills in order to design early intervention programs.

Regarding GMSs, 9.7% of children showed difficulties in performing these motor skills. This result corroborated with that found in the literature, since the studies indicated that the majority of children scored within or above the satisfactory classification for GMSs. An example of this was recorded in the studies carried out by Cook et al. [152] and Draper et al. [153,154], where more than 85% of the children showed no difficulties in GMSs. GMSs are fundamental to the development of social skills [155,156] and influence children’s PA levels and health [157,158]; as for FMSs, it is important to diagnose GMSs early in preschoolers.

Regarding PA, the majority of children participated in structured PA (72.6%); however, only 43.5% fulfilled the weekly PA recommendations. These data suggest that although children performed structured PA, this was not enough to meet the recommendations proposed for PA intensity, that is, at least 60 min of MVPAs daily [137].

Regarding BMI classification, a prevalence of 14.5% of children with pre-obesity and 14.5% with obesity (pre-obesity + obesity = 29%) was recorded. In 2021/2022, in Portugal, the prevalence of childhood pre-obesity was 18.4% and obesity 13.5% (pre-obesity + obesity = 31.9%) [159]. Therefore, the prevalence of pre-obesity and obesity recorded in this study does not differ much from that studied at the national level, with a variance of only 2.9%. It has been documented that pre-obese and obese children engage less in PA [160], contributing to negative outcomes in physical fitness [161], health [162] and motor skills [101,102] and can lead to impaired cognitive development [82]. Furthermore, parents and educators tend to be less likely to encourage obese children to practice PA based on perceptions that children with this typology will have limited physical abilities, which may contribute to triggering a cycle of PA avoidance [160]. As a response to these consequences, obesity prevention should be a priority in the preschool years, as these are essential for establishing an intervention on eating behavior and PA [163,164,165].

After analyzing and discussing the prevalence of the variables included here, it was found that the characteristics of the children in this study were within the values found in the literature, which leads to the conclusion that the characteristics of the children in these variables contributed to reducing a possible bias in this study.

### 4.2. Associations between Study Variables and Their Influence on Mathematical Performance

After studying the relationship between the prevalence of the study variables and those in the literature, the aim was to analyze the associations between all the variables and their direct and indirect influence on mathematical performance.

In the study on the association between the variables, in relation to FMSs, these were mainly influenced by VMI (VMI-AD, r = 0.975; *p* < 0.01) with a small contribution from FMC (TBT-AD, r = 0.177; *p* > 0.05), with no association between VMI skills and FMC. Although VMI and FMC contributed to FMS performance, VMI made a significantly higher contribution than FMC. When studying the relationship between VMI and FMC, no association was found between these specific motor skills. However, studies have demonstrated an interrelationship between VMI and FMC [166], where bead threading tasks have been associated with shape copying tasks [167], with FMC being a precursor to VMI [168]. Unlike FMSs, GMSs were strongly influenced by all the tests that make it up (aiming and catching r = 0.765; balance r = 0.716). However, among the GMS tests, the association was trivial (aiming and catching and balance: r = 0.099). Thus, the propulsive and repulsive manual skills assessed by the aiming and catching tests do not contribute to balance and vice versa.

Regarding the association between FMSs and GMSs, there was a small positive association between these motor skills (r = 0.224; *p* > 0.05). However, this result was influenced by VMI, since there was only one positive and significant association between GMSs and VMI (r = 0.269; *p* < 0.05). This result suggests that GMSs can positively influence FMSs by significantly increasing VMI skills. Although the literature shows a moderate positive relationship between FMSs and GMSs during child development, from r = 0.30 [169] to r = 0.60 [170], this is not consensual. Some studies have shown a moderate positive correlation between these skills in preschool children [53,171,172,173], and other studies have disagreed with the correlation [174,175,176]. These results can be justified, on one hand, by the fact that these studies assessed motor skills during short periods of age using very heterogeneous tasks [177] and, on the other hand, by the fact that motor skills do not follow linear developmental trajectories [178]. It has also been mentioned that in the same motor action it is difficult to clearly differentiate the involvement of each of the motor skills (GMSs and FMSs), since they coexist and are fundamental for the efficient performance of the task [178,179]. Regarding the association between the practice of structured PA and PA intensity with GMSs and FMSs, the associations found were small or trivial. It is worth highlighting the fact that SPAs had a small significant negative influence on FMC (TBT-AD, r = −0.267; *p* < 0.05) and a non-significant positive influence on VMI (VMI-AD, r = 0.217; *p* > 0.05). GMSs also had a small significant contribution from SPAs (r = 0.252; *p* < 0.05) and non-significant from LPAs (r = 0.204; *p* > 0.05). MVPAs did not contribute to the increase in either GMSs or FMSs. Thus, these results suggest that structured PA alone was not enough to influence motor skills and that it was SPAs that exerted the greatest influence, mainly through a decrease in FMC and a small increase in VMI and GMSs. In fact, most of the tasks associated with the development of VMI are graphomotor tasks that mainly require SPA. On the other hand, activities for the development of FMC require speed and precision (non-graphomotor) and may require an increase in PA intensity [180,181,182,183]. In the study carried out by Roth et al. [184], they concluded that SPA contributed to a decrease in motor skills. This decrease could be explained by the amount of time children spend watching TV or using their mobile phones or tablets [185,186]. It has also been reported that extensive tablet use affects FMC and VMI [187,188]. Although digital technologies can support self-directed learning in young children, they have negative consequences for the development of FMSs [189].

Although researchers agree that children in the first years of life should be very active through structured and unstructured PA [190,191], the results show that not all forms or configurations of PA are equally effective in promoting motor skills [84]. Motor skills, such as locomotion or object manipulation, are not acquired innately as children grow up [192]; they should be learned and practiced [34,35] through structured programs during early childhood to facilitate its acquisition [33]. However, many kindergartens focus their work mainly on academic content, reducing children’s opportunities to develop motor skills in these environments [155,193]. Since structured PA is considered more efficient for the development of GMSs and FMSs compared to unstructured PA [194,195,196], there is an urgent need to implement structured PA programs to prevent difficulties in motor skills [197,198,199]. During early childhood, children with difficulties in motor skills have a lower perception of their skills and consequently have more difficulties in engaging in more challenging PA [48], participate less in playing [200], and have more problems with their peers, potentially becoming victims of exclusion [201]. In this sense, despite the importance of the early development of motor skills in children, it has been neglected in educational practices [82], resulting in a growing increase in these difficulties in recent years [202,203]. Therefore, more important than the intensity of PA, it seems that structured PA is the most important prerequisite for the development of motor skills in early childhood.

Regarding the association between BMI, PA, and motor skills, it was found that the fact that children performed structured PA on a weekly basis and different levels of PA were not enough to influence BMI classification. Similarly, motor skills were not influenced by BMI classification either, with only a small, non-significant decrease in GMSs (r = −0.216; *p* > 0.05). Therefore, the results suggest that children’s adherence to the practice of structured PA, as well as the intensity of PA, had no effect on reducing overweight or obesity. This result can be explained by the fact that some parents enroll their children in structured PA when they reach levels of obesity above expectations. Supporting these results, studies concluded that being less physically active was associated with thinness, and not with overweight or obesity, and time spent in MVPAs was positively associated with BMI [204,205]. However, other studies have shown an inverse relationship between PA and BMI where overweight and obesity were associated with lower PA levels [206,207]. Regarding motor skills, the results indicated that the increase in BMI score did not influence performance of FMSs (FMS: r = 0.050; *p* > 0.05; FMC: r = 0.096; *p* > 0.05; VMI: r = −0.029; *p* > 0.05) and only contributed to a small decrease in GMSs (GMS: r = −0.216; *p* > 0.05; aiming and catching: r = −0.188; *p* > 0.05; balance: r = −0.130; *p* > 0.05). A similar result was recorded in the study carried out by Kakebeeke et al. [208] on 476 preschool children (3.9 ± 0.7 years). The authors concluded that body composition measurements (BMI, skin folds, and abdominal perimeter) were not related to FMSs and found only a small decrease in GMSs and only in children with a high percentage of fat mass.

Exclusively in relation to the association between obesity and FMS, the results in the literature do not suggest any association between FMSs, obesity, and BMI in preschool children [209]. Regarding the relationship between obesity and GMSs, the literature has shown that children who are overweight or obese have more difficulty in some GMSs, such as jumping, running, and balancing [210,211]. Other studies concluded that GMSs were not associated with childhood overweight and obesity [212,213,214]. However, a study carried out on obese children, following the intervention of a structured PA program, showed a reduction in average body weight and an increase in the performance of GMSs [215]. Other studies concluded that obesity leads to a decrease in PA [41] and contributes to a decrease in GMSs [38,39,40,102] and physical fitness and a consequent decrease in participation in structured PA programs, contributing to an increase in obesity [36]. In this sense, we are facing a vicious cycle of physical inactivity that promotes a decrease in motor skills and an increase in obesity, with all the associated risks.

One of the main aims of this research was to study the influence of FMSs, GMSs, PA, and BMI on mathematical performance assessed by the WPPSI-R. The results showed that only the VMI-AD test was positively and significantly associated with the WPPSI-R arithmetic test (Beta = 0.464; T = 4.055; *p* < 0.001), influencing it by 21.5 per cent (R^2^ = 0.215). Thus, the VMI-AD test can be used to explain the results of the WPPSI-R arithmetic test, with values close to a large effect (R > 0.26) [112]. By placing the VMI-AD test as the dependent variable in the MLR model, no variables were included in the model, suggesting that FMC, GMSs, PA, and BMI classification did not directly influence VMI-AD and therefore were not indirect predictors of the results of the WPPSI-R math test. These results support recent systematic reviews, which concluded that VMI was the specific motor skill that was most consistently positively and significantly associated with math performance [29,64]. Regarding the role played by PA in academic performance, a recent systematic review found no clear consensus on the effects of PA on academic results in early childhood [216]. However, some studies reinforce the idea that PA with high coordination demands improves higher cognitive processes, particularly executive function, contributing to children’s school learning [217,218], and that aerobic training with low coordination demands has no influence on academic performance [219,220,221]. Regarding GMSs, although some authors suggest significant associations between GMSs and math [22,26,97], the literature is inconsistent and insufficient to report the relationship between the specific GMS components and math performance in preschoolers [29,64,193,222]. Despite the inconsistency of the results, these skills should be part of the kindergarten teacher’s work, since this educational period should contribute to children’s all-round development [222]. Furthermore, preschool children involved in a structured, cognitively challenging GMS program could contribute to the improvement of mathematical skills through the direct effect exerted on the improvement of FMSs [223].

Regarding FMSs, its importance for school readiness has been suggested (Grissmer et al., 2010 [52]). In a systematic review carried out by Flores et al. [64], one of the objectives was to identify the specific motor skills that would most influence mathematical performance in preschool children with typical development; the authors concluded that both FMS and VMI were predictors of mathematical performance, with an advantage for VMI skills, which showed more robust and consistent results. However, children with better FMC may be better at manipulating objects, such as pencils or notebooks, which allows them to direct additional attentional resources to new learning, particularly VMI [224]. Thus, a child with a good FMC when performing an academic task may impose a lower cognitive load compared to a child who still has FMC difficulties [225,226].

After analyzing the associations and influences of the variables on mathematical performance, this study suggests that mathematical performance was not influenced by FMC, GMSs, PA, and BMI, and only VMI skills directly contributed to increasing mathematical performance in preschool children with typical development. However, since VMI skills were positively and significantly associated with GMSs (r = 0.269; *p* < 0.05), it seems that these skills could contribute to improving VMI skills. Thus, math skills are directly influenced by VMI, and since GMSs were positively and significantly associated with VMI, they could directly contribute to increasing VMI and thus indirectly math skills.

The direct and indirect relationships and influences presented in this study between motor skills, PA, and obesity on mathematical performance demonstrate that new policies should support kindergartens with the implementation of intervention programs at this level.

## 5. Conclusions

Understanding which specific competencies are related to the mathematical performance of preschool children has significant implications for the implementation of educational programs at this level to prevent gaps in subsequent years.

This study demonstrated that only VMI directly, positively, and significantly influenced the mathematical performance of typically developing preschool children. However, FMSs and GMSs may indirectly contribute to mathematical performance, as evidenced by their associations with VMI.

Typically, motor skills are not highly valued at this educational level, primarily due to the common misconception that children naturally develop motor skills during their developmental process. In this context, our results suggest the implementation of structured programs to develop VMI and thus enhance mathematical performance.

### Limitations

This study has several limitations that need to be considered. Firstly, it is essential to highlight that the sample was not representative of the population. This situation is primarily justified by the high number of assessments and data collection at different time points. Consequently, some children who were not eligible for the study either missed some assessments or did not provide necessary information for the research, leading to their exclusion. Another limitation is related to the instruments used, especially the tests for assessing FMSs. This was the first time these tests were used, and there were no existing data in the literature to discuss the results obtained here. More studies using the TBT-AD and VMI-AD are needed to increase the consistency of the conclusions.

Another limitation is associated with the acquisition of data on PA because, given the nature of the study, the type, frequency, and intensity of PA were obtained subjectively using the Pre-PAQ. It would likely be beneficial for this and future studies to include objective assessments of certain physical abilities in children. Regarding data collection on obesity levels, although BMI is the most commonly used measure for classifying overweight and obesity, other measures can be used in the preschool age, such as skinfold measurements or waist-to-hip ratio [188].

In future similar studies, it would be valuable to consider the socioeconomic status of the children, as preschool children in poverty are at a higher risk for lower performance in primary school and lower high school completion rates. Besides socioeconomic status, research has shown that children living with only one parent tend to be more physically active and spend more time playing outdoors. Future research may also consider this aspect.

## Figures and Tables

**Table 1 behavsci-13-01000-t001:** Students’ number distribution by age group, means and standard deviations.

Age Group	Male	Female	Total	Mean	Standard Deviation
3 years	3	5	8	3.08	0.03
4 years	12	4	16	4.08	0.14
5 years	14	19	33	5.06	0.03
6 years	3	2	5	6.02	0.01
Total	32	30	62	4.63	0.81

**Table 2 behavsci-13-01000-t002:** Distribution of the prevalence classification in the performance of the WPPSI-R arithmetic test.

Math Skills		
WPPSI-R	*N*	%
Much lower	2	3.2
Lower	13	21
Medium	36	58.1
Higher	9	14.5
Much higher	2	3.2

**Legend**: WPPSI-R—arithmetic test; *N*—number; %—percentage.

**Table 3 behavsci-13-01000-t003:** Distribution of the prevalence classification of children regarding FMS performance.

FMS								
FMC: TBT-AD			VMI: VMI-AD			Total		
Classification	*N*	%	Classification	*N*	%		*N*	%
Severe disorder	0	0	Very low	3	4.8	Lots of difficulties	0	0
Moderate disorder	21	33.9	Low	15	24.2	Some difficulties	7	11.3
Medium	23	37.1	Medium	41	66.1	Medium	24	67.7
Good	18	29	High	3	4.8	Good	13	21
Very good	0	0	Very high	0	0	Very good	0	0

**Legend**: FMS—fine motor skills; FMC—fine motor coordination; VMI—visuomotor integration; TBT-AD—adapted threading beads test; VMI-AD—adapted visuomotor integration test; N—number; %—percentage.

**Table 4 behavsci-13-01000-t004:** Distribution of the classification of the prevalence of children regarding the performance of GMSs.

GMS (MABC-2)										
			Balance						HMG	
Classification	Aiming and Catching		Static		Dynamic		Total			
	*N*	%	*N*	%	*N*	%	*N*	%	*N*	%
Motor disorder	6	9.7	1	1.6	1	1.6	1	1.6	0	0
Risk of motor disorder	2	3.2	6	9.7	4	6.5	4	6.5	6	9.7
Without risk	54	87.1	55	88.7	57	91.9	57	91.9	56	91.9

**Legend**: GMS—gross motor skills; MABC-2—Movement Assessment Battery for Children-2; N—number; %—percentage.

**Table 5 behavsci-13-01000-t005:** Practice of physical activity.

PA											
Weekly Structured PA						Fulfilment of Recommendations			Time Spent per Intensity		
Result	*N*	%	Days	*N*	%	Days	*N*	%	Intensities	M	Dp
No	17	27.4	0	17	27.4	0 days	8	12.9	Sedentary	328.95	155.23
			1	29	46.8	1 day	11	17.7	Slight	131.93	96.39
Yes	45	72.6	2	14	22.6	2 days	16	25.8	Moderate to vigorous	292.66	197.89
			3	2	3.2	3 days	27	43.5			

**Legend**: *N*—number; %—percentage; M—mean; Dp—standard deviation; PA—physical activity.

**Table 6 behavsci-13-01000-t006:** Mean and standard deviation of BMI and associated percentile and distribution of classification relative to BMI classification.

BMI							
	M	Dp	Maximum	Minimum	Classification	*N*	%
**Percentile**	59.62	32.16	99.80	0.10	Low weight	5	8.1
					Normal weight	39	62.9
**BMI**	16.37	2.22	24.50	11.12	Pre-obesity	9	14.5
					Obesity	9	14.5

**Legend**: *N*—number; %—percentage; BMI—body mass index; M—mean; Dp—standard deviation.

**Table 7 behavsci-13-01000-t007:** Association between all study variables.

	1	2	3	4	5	6	7	8	9	10	11	12	13	14	15	16
1. WPPSI-R	1	−0.014	0.464 **	0.454 **	0.084	−0.041	−0.134	−0.092	−0.001	0.018	0.038	−0.104	0.052	−0.009	−0.134	−0.032
2. TBT-AD		1	−0.045	0.177	−0.199	0.057	−0.108	−0.072	−0.186	−0.137	−0.064	−0.124	-0.267 *	−0.027	−0.018	0.096
3. VMI-AD			1	0.975 **	0.187	0.121	0.160	0.213	0.269 *	0.139	0.177	−0.084	0.217	0.096	−0.135	0.029
4. FMS				1	0.140	0.132	0.133	0.194	0.224	0.106	0.160	−0.110	0.154	0.089	−0.137	0.050
5. Aiming and catching					1	−0.059	0.189	0.099	0.765 **	−0.039	−0.195	−0.100	0.194	0.147	−0.104	−0.188
6. Static balance						1	0.156	0.759 **	0.450 **	−0.115	−0.137	−0.140	0.122	0.144	−0.116	−0.050
7. Dynamic balance							1	0.689 **	0.579 **	−0.049	−0.138	0.162	0.154	0.120	0.049	−0.022
8. Total balance								1	0.716 **	−0.118	−0.153	0.025	0.180	0.156	−0.057	−0.130
9. GMS									1	−0.103	−0.235	−0.054	0.252 *	0.204	−0.110	−0.216
10. Do structured PA										1	0.655 **	0.272 *	0.259 *	−0.171	0.193	−0.030
11. Time Structured PA											1	0.215	0.026	−0.161	0.153	0.040
12. Days that fulfil the recommendations												1	0.181	0.068	0.764 **	0.049
13. Time SPA													1	0.008	0.029	0.054
14. Time LPA														1	0.005	0.015
15. Time MVPA															1	0.003
16. BMI																1

**Legend**: WPPSI-R—arithmetic test; TBT-AD—adapted threading beads test; VMI-AD—adapted visuomotor integration test; FMS—fine motor skills; GMS—gross motor skills; PA—physical activity; SPA—sedentary physical activities; LPA—light physical activities; MVPA—moderate to vigorous physical activities; BMI—body mass index; **—the correlation is significant at the significance level < 0.01; *—the correlation is significant at the significance level < 0.05.

**Table 8 behavsci-13-01000-t008:** Summary of the MRLM according to the score obtained on the WPPSI-R (dependent variable).

MRL	R	R^2^	R^2^ (Aj)	R^2^-R^2^ (Aj)	F	*p*
1	0.464	0.215	0.202	0.013	16.447	0.000 *

**Legenda**: MLR—multiple linear regression model; MLR 1—*test score VMI-AD*; R—Pearson’s correlation of the WPPSI-R with the VMI-AD; R^2^—influence of the VMI-AD on the WPPSI-R; R^2^ (Aj)—R^2^ adjusted; R^2^-R^2^ (Aj)—explains the generalization of the model; F—F ratio (significance of R^2^); *p*—significance level; *—*p* < 0.001.

**Table 9 behavsci-13-01000-t009:** Table of MLR coefficients selected by the stepwise method.

Model	β	Sd	Beta	T	*p*
Constant	−0.409	2.548		−0.161	0.873
VMI-AD	0.115	0.028	0.464	4.055	0.000 *

**Legend**: β—non-standardized coefficients; Sd—standard deviation; Beta—standardized coefficients; T—test T; *p*—significance level; *—*p* < 0.001.

**Table 10 behavsci-13-01000-t010:** Tests excluded from MLR.

Testes	Beta in	T	*p*	*r*
TBT-AD	0.007	0.063	0.950	0.008
FMS	0.033	0.063	0.950	0.008
Aiming and catching	−0.003	−0.024	0.981	−0.003
Static balance	−0.098	−0.852	0.398	−0.110
Dynamic balance	−0.213	−1.878	0.065	−0.238
Total balance	−0.200	−1.736	0.088	−0.220
GMS	−0.200	−1.142	0.258	−0.147
Do structured PA	−0.047	−0.404	0.688	−0.052
Number of structured PA performed	−0.058	−0.500	0.619	−0.065
Number of days fulfilling recommendations	−0.065	−0.566	0.573	−0.074
Compliance with weekly recommendations	−0.066	−0.568	0.572	−0.074
Total time spent in SPA	−0.05	−0.428	0.671	−0.056
Total time spent on LPA	−0.054	−0.468	0.642	−0.061
Total time spent on MVPA	−0.072	−0.625	0.535	−0.081
BMI classification	−0.046	−0.395	0.694	−0.051

**Legend**: TBT-AD—adapted threading beads test; FMS—fine motor skills; GMS—gross motor skills; PA—physical activity; SPA—sedentary physical activities; LPA—light physical activity: MVPA—moderate or vigorous physical activity; BMI—body mass index; T—Test T; *p*—significance level; *r*—Pearson’s correlation with the WPPSI-R.

## Data Availability

Data are available upon request from the corresponding author.

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
