# Peer review of "Relationships between Math Skills, Motor Skills, Physical Activity, and Obesity in Typically Developing Preschool Children"

_behavsci, 2023, doi:10.3390/bs13121000_

Round 1

Reviewer 1 Report

Comments and Suggestions for Authors

REVIEWER Comments:

I appreciate the time and effort the authors have dedicated to this work. The submitted manuscript offers valuable insights into the relationship between physical activity (PA), motor skills, and early childhood development. While the introduction aptly outlines the study's significance, it would benefit from a clearer delineation of the literature gap addressed. The methods section is robust but would gain from elaborating on participant selection criteria and the rationale for chosen tools. The discussion is comprehensive, but it could be structured more coherently with subheadings and would benefit from a deeper exploration of contradictions and discrepancies with existing literature. Additionally, emphasis on the study's implications, its limitations, and potential future research directions is recommended. Overall, this manuscript is a noteworthy contribution to the field; however, refinements in presentation and clarity will amplify its impact.

Introduction

The introduction covers a broad range of topics, from motor skills and their types to physical activity, obesity, and mathematical performance. Consider restructuring the introduction for a more logical and clear flow. Begin with the general importance of early childhood development, delve into motor skills and their importance, followed by physical activity, and then obesity and its relationship with PA and motor skills.

1.      There seems to be an inconsistency in the abbreviation for motor skills. It's abbreviated as "MS" in line 103 and later "PA" is defined as "Physical Activity". It's essential to maintain consistency with abbreviations throughout.

2.     Transition between motor skills (GMS and FMS) and physical activity could be smoother. Consider providing a more direct link between the development of these skills and the role of physical activity.

3.      The text would benefit from a thorough proofreading. Some sentences are quite lengthy, making them a bit cumbersome to read. Consider breaking them up for clarity.

4.      The secondary objective (lines 146-149) seems slightly detached from the primary objective. Consider providing a clearer rationale for its inclusion. What is the significance of assessing the consistency of studied variables with existing literature?

Methods

1.      Some of the instructions provided seem too prescriptive for a Methods section (e.g., "However, the administrator should not say 'Good' or 'Correct' after a correct response"). It might be beneficial to condense these while ensuring the main essence is retained.

2.      Consider providing a brief rationale for choosing only the Arithmetic test from the Verbal subtest of the WPPSI-R for this study.

Discussion:

1.      The Discussion section is comprehensive, thorough, and discusses the results in light of previous research. The flow of ideas is systematic. However, the extensive length of the section may be overwhelming to some readers.

2.      It is commendable that the authors have incorporated a wide array of references to support their arguments. However, it may be helpful to more succinctly summarize the range of results in the existing literature to provide a more concise perspective.

3.      The implications of FMS and GMS on learning outcomes and the importance of early diagnosis are well-stated. However, the authors might want to consider expanding on the potential interventions or strategies for children who have been identified with FMS or GMS difficulties.Conclusion:

4.      The section highlights a complex relationship between different types of physical activity (structured, unstructured, MVPA, etc.) and motor skills. To avoid confusion, consider clarifying terminology or providing a brief definition when first introducing these terms.

Author Response

Response to Reviewer 1 Comments

Dear reviewer, we thank you for your work on this review. Your observations and suggestions were very pertinent in improving this article. In this sense, we will try to answer all the proposed corrections.

Thank you very much.

Introduction

The introduction covers a broad range of topics, from motor skills and their types to physical activity, obesity, and mathematical performance. Consider restructuring the introduction for a more logical and clear flow. Begin with the general importance of early childhood development, delve into motor skills and their importance, followed by physical activity, and then obesity and its relationship with PA and motor skills.

Dear reviewer, thank you for your comments and your work. You have read this article very carefully and with great scientific rigour and quality. In fact, this introduction covers many concepts that are directly and indirectly associated with the holistic development of pre-school children.

We have followed a very similar introduction logic to the one you suggest. In this sense, we begin the introduction with the importance of global development in early childhood, and the school as a fundamental place for intervention. Afterwards, we justify the importance of motor skills in children's overall development, and the practice of physical activity as one of the factors that contributes most, not only to the development of motor skills, but also to health, particularly in controlling obesity.

Next, as the aim of this article is by no means exclusively aimed at physical education and sports professionals, but also at others interested in the area, namely nursery school teachers, it was necessary to define the concepts in some depth, and not just refer to the references.

Subsequently, and in line with the aim of the article, we explained the advantages of including maths and its relationship with motor skills, and how these motor skills could be influenced by physical activity, as well as the relationship between physical inactivity and obesity and vice versa. We don't intend to go into too much detail in the introduction about the relationships between all the variables, as this has been extensively covered in the discussion.

Finally, the objectives were defined, and we would like to thank you in advance for your strong contribution to the reformulation of the objectives.

1.      There seems to be an inconsistency in the abbreviation for motor skills. It's abbreviated as "MS" in line 103 and later "PA" is defined as "Physical Activity". It's essential to maintain consistency with abbreviations throughout.

Thank you very much for your comment. In fact, no matter how many corrections are made, there are always some flaws at this level. Although it appears many times in the text, and in order not to have too many acronyms, it was our intention not to put the acronym for motor skills, but rather in its components, fine motor skills (FMS), gross motor skills (GMS), fine motor coordination (FMC) and visuomotor integration (VMI). We have already made this correction, removed (MS) and changed the concept of physical activity to PA throughout the article.

2.     Transition between motor skills (GMS and FMS) and physical activity could be smoother. Consider providing a more direct link between the development of these skills and the role of physical activity.

You're absolutely right. In fact, there seems to be a break in the sequence of the text. In this sense, we consider it important to add, following on from a previous sentence, the implication between the development of motor skills and the increase in children's involvement in PA practice. We chose not to elaborate further on the direct relationship between the concepts since it has already been extensively documented in the discussion.

We have therefore added the following sentence to link the concepts: "Since the development of motor skills in early childhood contributes to a greater involvement of children in the practice of PA, it is important to characterise and classify this concept."

3.      The text would benefit from a thorough proofreading. Some sentences are quite lengthy, making them a bit cumbersome to read. Consider breaking them up for clarity.

Thank you for your suggestion. We realise that sometimes, due to the number of variables involved, longer sentences appear in an attempt to relate the concepts. However, since it is essential that the reader enjoys a clear and objective reading, we have considered shortening some sentences, dividing some into paragraphs and removing some extra content to make the text clearer and more accessible.

4.      The secondary objective (lines 146-149) seems slightly detached from the primary objective. Consider providing a clearer rationale for its inclusion. What is the significance of assessing the consistency of studied variables with existing literature?

Thanks again for the suggestion. In fact, this study does not have a single distinct objective. Rather, it has three objectives that complement each other. These objectives are clearly evident in the results and discussion. We are therefore considering rewording the paragraph:

“In view of the above, given that pre-school age is a sensitive period for the development of motor skills and that low levels of PA in this period can compromise their development and contribute to weight gain and obesity, the following objectives were outlined: 1) To analyse the prevalence of difficulties shown by the children in mathematical skills, motor skills, compliance with weekly PA recommendations and obesity levels, comparing the results with those in the literature, with the aim of justifying the fit or bias of the study variables; 2) To study the associations between all the variables with the aim of analysing the direct relationships between them; 3) To analyse the direct influence of motor skills, PA levels and obesity on mathematical skills.”

Methods

1.      Some of the instructions provided seem too prescriptive for a Methods section (e.g., "However, the administrator should not say 'Good' or 'Correct' after a correct response"). It might be beneficial to condense these while ensuring the main essence is retained.

Thank you for your comment. In fact, this observation was shared by the other reviewers. In this sense, we have removed a lot of descriptive information from the procedures for applying the instruments and referred to the references.

2.      Consider providing a brief rationale for choosing only the Arithmetic test from the Verbal subtest of the WPPSI-R for this study.

Thank you for your suggestion. In this regard, we have included the following sentence: "Given the scarcity of instruments for assessing mathematical skills in pre-school children in our country, this scale was chosen because it is validated for the population in question, it is the most widely used in national research at this level, it does not require much training and it is easy to apply."

Discussion

1.    The Discussion section is comprehensive, thorough, and discusses the results in light of previous research. The flow of ideas is systematic. However, the extensive length of the section may be overwhelming to some readers.

Thank you for your comment. This was also a problem we identified. In fact, the number of variables involved in the study required a great deal of research into the prevalence and relationships between them, so the discussion became very long. This was the only way we could justify all the proposed objectives. In this sense, we opted for a robust discussion in view of the immense literature on these lines of enquiry (mathematical skills, motor skills, physical activity and obesity). As a first attempt, we divided the discussion into several sub-chapters to make it easier to read, but, as you say, we lost the flow of ideas and their connection. So we divided the discussion into just 2 subchapters, one related to the prevalence of the variables and the other to the associations between them and their influence on maths performance.

2.              It is commendable that the authors have incorporated a wide array of references to support their arguments. However, it may be helpful to more succinctly summarize the range of results in the existing literature to provide a more concise perspective.

When we finished the article, we came across the extensive bibliography. We still removed many authors because we realised that the bibliographical references are very extensive. However, given the scale of the variables investigated, we did not consider it ethically correct to remove other authors due to their contribution to the different areas of research. When we analysed the number of prevalences and relationships studied in this research, as well as the amount of robust literature that exists in these areas, we were fully aware that this work would have to be supported by a wide range of quality references.

3.      The implications of FMS and GMS on learning outcomes and the importance of early diagnosis are well-stated. However, the authors might want to consider expanding on the potential interventions or strategies for children who have been identified with FMS or GMS difficulties.Conclusion:

Dear reviewer, thank you once again for your contributions. This research project is very recent (starting in 2022). It aims to be transversal to various types of children. We are at an embryonic stage of the project. The tests to assess fine motor skills associated with mathematical skills have been adapted by us for this purpose and require further validation (Flores, P.; Coelho, E.; Mourão-Carvalhal, M.I.; Forte, P. M. Preliminary Adaptation of Motor Tests to Evaluate Fine Motor Skills Associated with Mathematical Skills in Preschoolers. Eur. J. Investig. Health Psychol. Educ. 2023, 13, 1330-1361. https://doi.org/10.3390/ejihpe13070098)

This project will end in 2026, and one of the aims will be to study children with different cognitive and motor difficulties. In this sense, given the aims of the article, it would be very premature to add or indicate interventions for children with difficulties in FMS or GMS.

4.      The section highlights a complex relationship between different types of physical activity (structured, unstructured, MVPA, etc.) and motor skills. To avoid confusion, consider clarifying terminology or providing a brief definition when first introducing these terms.

Once again we thank you for your fantastic contributions. In the discussion, we decided not to define these concepts again, as we did in the introduction. In the introduction, when we referred to calorific expenditure as a function of physical activity intensity, we broke down the metabolic equivalent (MET) correspondence to physical activity intensity levels (sedentary, light, moderate and vigorous). Similarly, in the introduction, when classifying the types of physical activity, we distinguished between structured physical activity, which refers to formal physical activity carried out under the direct guidance and instruction of a specialist for this purpose, and unstructured physical activity, which refers to non-formal physical activity carried out spontaneously and freely by children in the form of play. The components of motor skills were also defined and contextualised in the introduction.

We emphasise the relevance of this observation, as these concepts could have been developed in the discussion, but we chose to do so in the introduction. If they had been developed in the discussion, it would have been much longer and heavier for the reader.

Dear reviewer, we would like to thank you for your excellent work on this review. Your contributions have been essential in substantially improving this article. Thank you for the rigor and quality of your suggestions.

Best regards

Reviewer 2 Report

Comments and Suggestions for Authors

Dear Authors.

First of all, I would like to congratulate you for the work done, which certainly seems very interesting. I would like to make some remarks that I believe could lead to an improvement of the proposal.

Abstract: several variables are presented, but only notions of the relationship between fine motor skills and mathematical skills are left. I believe that all variables should be treated in the same way.

Introduction: I believe that some concepts that, theoretically, should be known, to a greater or lesser extent, by a reader interested in a work of these characteristics, are overdeveloped.

2.1. Sample: if no problems arise from this circumstance (premature birth), how can we justify not having this number of participants?

2.2. Instruments: I believe that this is overdeveloped. It is necessary for the reader to have notions about the instrument, but many details can be avoided that make the reading dense and, really, not very useful.
I still appreciate an excess of information that is not relevant and, on the other hand, if the instrument is not validated for the Portuguese population (TBT-AD), how was this done? This should be considered for all the variables considered in the work.

Scoring and Classification of Fine Motor Skills: where does this classification come from?

2.2.4. Obesity diagnosis: I believe that, to be more precise, BMI is used to assess body composition and, based on the results found, it could refer to underweight, normal weight, overweight or obesity.

2.3. Procedures: I find reiterations, insofar as there are aspects that have already been mentioned. Perhaps, here, only the order and little else would be appropriate, but not to repeat. Once again, there is an excess of information.

I think it would be relevant not to incorporate 3, 4 or 5 references that mention the same aspect, since it has resulted in 10 pages of references.

Greetings.

Author Response

Response to Reviewer 2 Comments

Dear reviewer, we thank you for your work on this review. Your observations and suggestions were very pertinent in improving this article. In this sense, we will try to answer all the proposed corrections.

Thank you very much.

1.              Abstract: several variables are presented, but only notions of the relationship between fine motor skills and mathematical skills are left. I believe that all variables should be treated in the same way.

In fact, you're absolutely right. It didn't make sense not to name the variables studied in the results. In this sense, it was written in the abstract: "The results indicated that only the fine motor skills of visuomotor integration were inserted into the multiple linear regression model (F < 0.001; r = 0.464; R2 = 0.215; p < 0.001), with gross motor skills, levels of physical activity and levels of obesity being excluded. Thus, it was concluded that mathematical skills were only directly and significantly influenced by visuomotor integration. However, visuomotor integration was positively and significantly associated with gross motor skills (r = 0.269; p < 0.05), and not with levels of physical activity and obesity."

2. 2.1. Sample: if no problems arise from this circumstance (premature birth), how can we justify not having this number of participants?

Thank you very much for your comment. When we started this research, we really wondered whether or not to include this type of child. We chose not to include them because a significant body of literature has shown that typically these children's most common problems are related to deficits in both fine and gross motor skills, which is justified in the article. In this sense, children with this typology could bias some results.

3. 2.2. Instruments: I believe that this is overdeveloped. It is necessary for the reader to have notions about the instrument, but many details can be avoided that make the reading dense and, really, not very useful. I still appreciate an excess of information that is not relevant and, on the other hand, if the instrument is not validated for the Portuguese population (TBT-AD), how was this done? This should be considered for all the variables considered in the work.

We clearly agree with the suggestion. This suggestion was echoed by the other reviewers. In fact, it is not necessary to exhaustively detail the instruments, administration procedures and materials to be used. We have therefore significantly shortened this point.

Regarding the tests to evaluate fine motor skills (TBT-AD and VMI-AD), these tests were validated for the Portuguese population in a study carried out by Flores et al., 2023 (Flores, P.; Coelho, E.; Mourão-Carvalhal, M.I.; Forte, P. M. Preliminary Adaptation of Motor Tests to Evaluate Fine Motor Skills Associated with Mathematical Skills in Preschoolers. Eur. J. Investig. Health Psychol. Educ. 2023, 13, 1330-1361. https://doi.org/10.3390/ejihpe13070098)

4. 2.2.4. Obesity diagnosis: I believe that, to be more precise, BMI is used to assess body composition and, based on the results found, it could refer to underweight, normal weight, overweight or obesity.

Thanks again for the suggestion. In fact, the literature associates BMI more with body composition. We have therefore replaced “The body mass index (BMI) was the selected anthropometric indicator for assessing obesity, …” to “The body mass index (BMI) was the selected anthropometric indicator for assessing body composition, …”

5. 2.3. Procedures: I find reiterations, insofar as there are aspects that have already been mentioned. Perhaps, here, only the order and little else would be appropriate, but not to repeat. Once again, there is an excess of information.

I think it would be relevant not to incorporate 3, 4 or 5 references that mention the same aspect, since it has resulted in 10 pages of references.

Thank you for your suggestion. We consider it unnecessary to mention the purpose of the study again at this point, which has been removed. With regard to the other assessments, we have also removed the names of the tests so as not to repeat implicit information. We have kept what is necessary and fundamental at this point for a correct understanding of the study design. In this sense, we present the methodology for applying the tests, collecting and analyzing the results chronologically and temporally.

The bibliographical references are very extensive. We considered keeping all the references for the following reasons: 1) the large number of variables studied and associated with each other presents a vast and high quality body of literature, which we would like to keep; 2) these references give more consistency to our study; 3) the fact that the literature differs on some points increases the discussion about them, and as a consequence more studies of high quality and evidence.

Dear reviewer, we would like to thank you for your excellent work on this review. Your contributions have been essential in substantially improving this article. Thank you for the rigor and quality of your suggestions.

Best regards

Reviewer 3 Report

Comments and Suggestions for Authors

Dear authors, 

The manuscript which aims to  analyse the relation-16 ships between math skills, motor skills, levels of physical activity, and obesity is very interesting. Please read what I suggest below, in case it might be of interest to you. 

Abstract

Too long. Please try to reduce its content.

2.1 Sample

Page 4, line 168. Table 1. How many males and females are in each group? Is the information necessary in view of the fact that no reference is made subsequently to the different tests according to the different age groups?

2.2 Instruments

Excessively long and cumbersome. It is suggested to reduce the information, relying on references to make it easier to read.

4. Discussion

Excessively long. It is suggested to revise and be more concise, focusing on a descriptive and comparative analysis, adding some reflection on it. 

Author Response

Response to Reviewer 3 Comments

Dear reviewer, we thank you for your work on this review. Your observations and suggestions were very pertinent in improving this article. In this sense, we will try to answer all the proposed corrections.

Thank you very much.

1. Abstract: Too long. Please try to reduce its content.

Thank you very much for your comment. It was indeed possible to reduce its content by 7 lines. We've only kept what's essential for the reader. Thank you.

2. 2.1 Sample: Page 4, line 168. Table 1. How many males and females are in each group? Is the information necessary in view of the fact that no reference is made subsequently to the different tests according to the different age groups?

Once again, we thank you for your contribution. Therefore, the complete data on the number of boys and girls by age group has been inserted into the table (3 years = 3 boys and 5 girls; 4 years = 12 boys and 4 girls; 5 years = 14 boys and 19 girls; 6 years = 3 boys and 2 girls).

3. 2.2 Instruments: Excessively long and cumbersome. It is suggested to reduce the information, relying on references to make it easier to read.

We clearly agree with the suggestion. This suggestion was echoed by the other reviewers. In fact, it is not necessary to exhaustively detail the instruments, administration procedures and materials to be used. We have therefore significantly shortened this point.

4. 4. Discussion: Excessively long. It is suggested to revise and be more concise, focusing on a descriptive and comparative analysis, adding some reflection on it. 

Once again, this suggestion was echoed by the other reviewers. In this sense, we have only discussed the essentials, i.e. we have focused on a discussion and reflection oriented solely towards the objectives of this study.

Dear reviewer, we would like to thank you for your excellent work on this review. Your contributions have been essential in substantially improving this article. Thank you for the rigor and quality of your suggestions.

Best regards

Round 2

Reviewer 1 Report

Comments and Suggestions for Authors

REVIEWER Comments:

I would like to express my appreciation for the authors' efforts in addressing the concerns and suggestions raised in my initial review.

The revisions made to the manuscript have significantly improved its clarity, depth, and overall quality. The authors have diligently responded to each point, demonstrating a thorough understanding of the subject matter and a commitment to scientific rigor. The changes implemented have effectively addressed the previously identified issues, enhancing the manuscript's contribution to the field.

In light of the revisions made, I am pleased with the current state of the manuscript and believe it is now ready for publication. The study's findings, which clearly present the impact of both fine and gross motor skills, as well as physical activity, on mathematical abilities and obesity in young children, are a valuable contribution to the existing body of literature.

Thank you for the opportunity to review this work. I commend the authors for their hard work and dedication to improving their manuscript.

Author Response

Thank you for your recommendations. We have made every effort to improve the manuscript.